# Integrated analysis of the transcriptome and metabolome of purple and green leaves of *Tetrastigma hemsleyanum* reveals gene expression patterns involved in anthocyanin biosynthesis

Jianli Yan[1], Lihua Qian[1], Weidong Zhu[2], Jieren Qiu[1], Qiujun Lu[3], Xianbo Wang[1], Qifeng Wu[4], Songlin Ruan[1]*, Yuqing Huang[1,5]*

1 Laboratory of Plant Molecular Biology & Proteomics, Institute of Biotechnology, Hangzhou Academy of Agricultural Sciences, Hangzhou, China, 2 Quzhou Academy of Agricultural Sciences, Quzhou, China, 3 Agricultural Science and Technology Education Terminal, Hangzhou Agricultural and Rural Bureau, Hangzhou, China, 4 Agriculture and Forestry Technology Promotion Center, Hangzhou Linan Agricultural and Rural Bureau, Hangzhou, China, 5 College of Agriculture and Biotechnology, Zhejiang University, Hangzhou, China

* ruansl1@hotmail.com(SR); huangyq@zju.edu.cn(YH)

## Abstract

To gain better insight into the regulatory networks of anthocyanin biosynthesis, an integrated analysis of the metabolome and transcriptome in purple and green leaves of *Tetrastigma hemsleyanum* was conducted. Transcript and metabolite profiles were archived by RNA-sequencing data analysis and LC-ESI-MS/MS, respectively. There were 209 metabolites and 4211 transcripts that were differentially expressed between purple and green leaves. Correlation tests of anthocyanin contents and transcriptional changes showed 141 significant correlations (Pearson correlation coefficient >0.8) between 16 compounds and 14 transcripts involved in the anthocyanin biosynthesis pathway. Some novel genes and metabolites were discovered as potential candidate targets for the improvement of anthocyanin content and superior cultivars.

## Introduction

*Tetrastigma hemsleyanum Diels et Gilg* (*T. hemsleyanum*, belonging to the family Vitaceae), also known as "Sanyeqing" (SYQ), is distributed in tropical to subtropical areas in Asia, mainly in southern China, such as Zhejiang, Guizhou and Guangxi provinces. The entire herb and its root tubers have been used as a broad-spectrum antibiotic material for the treatment of fever and sore throat in China for a long time. Previous findings have demonstrated that the principal and functional components of SYQ, such as polysaccharides and polyphenols are beneficial for health [1,2].

Flavonoids such as anthocyanins and flavanols are the most dominant and important components in SYQ [3,4]. These compounds are beneficial for human health and are an important

**Data Availability Statement:** All relevant data are within the paper and its Supporting Information files.

**Funding:** This work was sponsored by grants Active Design Project of Hangzhou Agricultural and Social Development Scientific Research (No. 20190101A05), and Science and Technology Innovation and Demonstration extension Fund of Hangzhou Academy of Agricultural Sciences (No. 2019HNCT-31).

**Competing interests:** The authors have declared that no competing interests exist.

group of pigments that colour the leaves and flowers of many plants [5]. Anthocyanins are also involved in biotic and abiotic stress defence responses [6,7]. It is vital to elucidate the flavonoid biosynthesis and regulatory pathways in SYQ. Although several genes in the anthocyanin regulation pathway have been identified [8], the unique regulation mechanism in SYQ remains unclear.

The colour of plant leaves are mainly determined by the composition and concentration of anthocyanins [9], products of the branched flavonoid biosynthesis pathway. The accumulation of anthocyanins is regulated by the expression of multiple genes, including chalcone synthase (CHS), chalcone isomerase (CHI), flavanone 3-hydroxylase (F3H), flavonoid 3′-hydroxylase (F3′H), flavonoid 3′,5′-hydroxylase (F3′5′H), dihydroflavonol 4-reductase (DFR), anthocyanidin synthase (ANS) and flavonoid 3-O-glucosyltransferase (UFGT)[10].

In the past decade, RNA sequencing has rapidly become an efficient approach to analyse the function of genes in a high-throughput way [11]. Transcripts with low abundance and unknown transcripts cannot be identified [12]. Liquid chromatography / mass spectrometry (LC/MS) has advanced metabolomics by enabling the discovery ofa large number of compounds compared with traditional chemical analysis [13,14]. Recently, the integrated analysis of metabolic and gene expression has been widely used in the exploitation network and correlation between metabolites and genes [15–17].

In this study, we explored the regulatory networks of flavonoid and anthocyanin biosynthesis in green (RG) and purple leaves (PL) of SYQ at the transcriptome and metabolome levels. The accumulation of different types of anthocyanin and the expression of related genes were investigated. A connection network was constructed to highlight the regulatory genes associated with specific metabolites. Our findings provide insights into the accumulation mechanism of SYQ leaf colour pigments and the regulation of anthocyanin biosynthesis.

## Materials and methods

### Plant materials and growth conditions

The green (RG) and purple leaf (PL) genotypes (S1 Fig) of *Tetrastigma hemsleyanum* (Sanyeqing) were collected from Guangxi and Hunan provinces, respectively. They were grown in the plant garden of Hangzhou Academy of Agricultural Sciences (Hangzhou, Zhejiang Province, China). Leaves at the third node away from the top were collected, frozen in liquid nitrogen and stored at -80˚C for RNA and metabolite isolation.

### RNA isolation and transcriptome sequencing

Total RNA was isolated with TRIzol reagent (Invitrogen, USA) according to the manufacturer's protocol. The transcriptome sequence library was constructed using NEBNext Ultra RNA Library Prep Kits for Illumina (NEB, USA). Sequencing was performed on an Illumina HiSeq 2500 platform (Novogene, China). The reads were aligned to the genome using TopHat (2.0.9) software after removing the reads containing adapter or poly-N and low-quality reads from the raw data. Total number of reads per kilobase per million reads (RPKM) of each gene was calculated based on the length of the gene and the counts of reads mapped to this gene. GO annotation was implemented using Blast2GO software. KOBAS (2.0) software was used for KEGG enrichment analysis of differentially expressed genes.

### Metabolic profiling

The samples were freeze-dried and crushed into power for metabolite isolation. One hundred milligrams of each sample were extracted with 1.0 ml 70% aqueous methanol at 4˚C overnight.

The extracts were absorbed (CNWBOND Carbon-GCB SPE Cartridge, 250 mg, 3 ml; ANPEL, Shanghai, China, www.anpel.com.cn/cnw) and filtered (SCAA-104, 0.22μm pore size; ANPEL, Shanghai, China, http://www.anpel.com.cn/) before LC-MS analysis.

The sample extracts were analysed using an LC-ESI-MS/MS system (HPLC, Shim-pack UFLC SHIMADZU CBM30A system, www.shimadzu.com.cn; MS, Applied Biosystems 6500 Q TRAP, www.appliedbiosystems.com.cn). A Waters ACQUITY UPLC HSS T3 C18 (1.8 μm, 2.1 mm*100 mm) was used for compound separation. The analytical conditions were set as follows: solvent system, water (0.04% acetic acid): acetonitrile (0.04% acetic acid); gradient program,95:5V/V at 0min, 5:95V/V at 11.0min, 5:95V/V at 12.0min, 95:5V/V at 12.1min, 95:5V/V at 15.0 min; flow rate, 0.40 ml/min; temperature, 40˚C; injection volume, 2 μl. Mass data acquisition was performed in both positive and negative modes using the following parameters: ion source, turbo spray; source temperature, 500˚C; ion spray voltage (IS), 5500 V; ion source gas I (GSI), gas II(GSII), and curtain gas (CUR) of 55, 60, and 25.0 psi, respectively; collision gas(CAD) was high. The mass fragmentations were compared to the HMDB (http://www.hmdb.ca), METLIN (http://metlin.scrippps.edu) and KEGG (http://kegg.jp) databases. The obtained data were used by SIMCA-P V12.0.0 Demo (Umetric, Umea, Sweden) for principal components analysis (PCA) and partial least-squares discriminant analysis (PLS-DA).

## Verification of candidate genes by quantitative real-time PCR (qRT-PCR)

The expression levels of transcripts involved in anthocyanin biosynthesis pathways were validated by qPCR using the same RNA samples used for sequencing. The primers were designed using beacon designer 7.8 software, and cDNAs were synthesized using SuperScript™ III First-Strand Synthesis SuperMix for qRT-PCR (Invitrogen). Power SYBR® Green PCR Master Mix (Applied Biosystems) was selected for the identification of the PCR products on a CFX384 Real-time PCR system (Bio-Rad). Three replicates were performed for each sample, and the primers of each sample are listed in S1 Table.

## Results

### Metabolic differences in green leaves and purple leaves

To compare the metabolite composition in the two accessions, LC-MS was used for metabolite identification. In total, 597 metabolites with known structures were detected in RG and PL. Differentially expressed proteins were defined as those with a variable importance for projection (VIP) value >1.5 compared to RG (Fig 1). Among the detected metabolites, 111 metabolites were upregulated and 98 metabolites were downregulated. The top 10 most differentially accumulated metabolites are listed in Table 1. The top 10 upregulated metabolites were mainly anthocyanins and flavone C-glycosides. Compared to RG, cyanidin 3-O-rutinoside showed the highest fold change in PL, with a log2 fold change (FC) (PL/RG) value of 20.59. Of the 10 most downregulated metabolites, quinic acid O-glucuronic acid had the lowest log2 fold change (FC) (PL/RG) value of -21.73.

All differentially accumulated metabolites were subjected to KEGG analysis (Fig 2).Metabolites participating in flavonoid biosynthesis, flavone and flavanol biosynthesis, tryptophan metabolism and biosynthesis of alkaloids derived from the shikimate pathway were predominantly enriched.

Principal component analysis (PCA) was employed to identify the differences in metabolite profiles among samples (Fig 3). The results showed that the first principal component (PC1, 52.98% of the total variables) was clearly separated in the RG and PL samples, indicating that the accumulation patterns of metabolites were different in green and purple leaves.

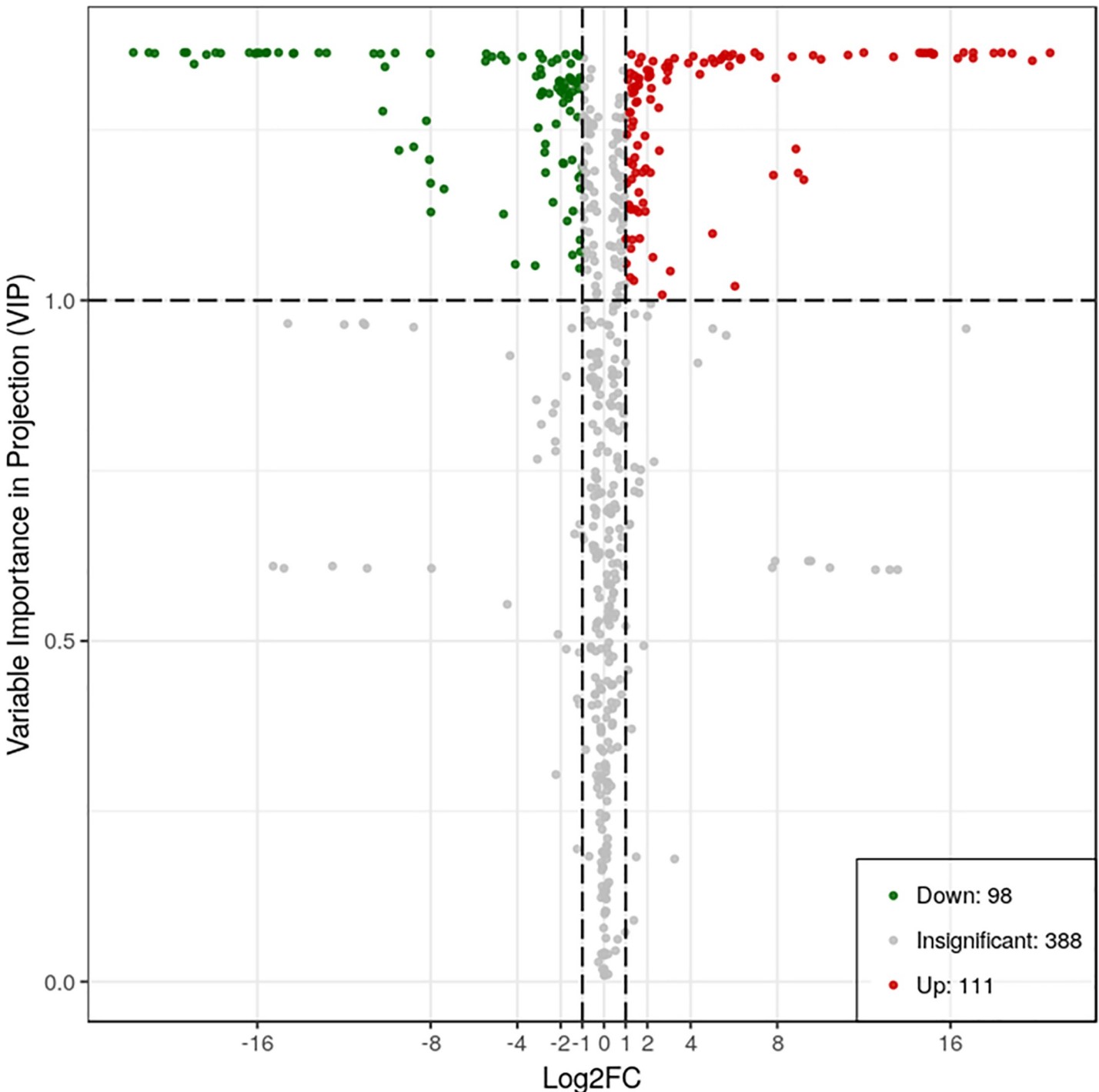

**Fig 1. VIP (variable importance for projection) plot of differentially accumulated metabolites in PL vs RG.** Red dots are upregulated metabolites, and green dots are downregulated metabolites.

### Transcriptome analysis in green leaves and purple leaves

After removing the low-quality reads, approximately 30,000,000 reads of each sample were generated. De novo transcriptome assembly using Trinity built 205,387 transcripts and

**Table 1. List of top 10 up- and downregulation metabolites in PL vs RG.** Differentially expressed proteins were defined as those with a variable importance for projection (VIP) value >1.5.

| Index | Compounds | Class | RG1 | RG2 | RG3 | PL1 | PL2 | PL3 | VIP | Fold_Change | Log2FC |
|---|---|---|---|---|---|---|---|---|---|---|---|
| pme1773 | Cyanidin 3-O-rutinoside (Keracyanin) | Anthocyanins | 9 | 9 | 9 | 19500000 | 11500000 | 11700000 | 1.363135 | 1581481.481 | 20.59285 |
| pmb0675 | C-pentosyl-apigenin O-p-coumaroylhexoside | Flavone C-glycosides | 9 | 9 | 9 | 4880000 | 18500000 | 819000 | 1.351613 | 896259.2593 | 19.77356 |
| pme3038 | 5-oxoproline | Amino acid derivatives | 9 | 9 | 9 | 2880000 | 7300000 | 2470000 | 1.362247 | 468518.5185 | 18.83775 |
| pme1793 | Pelargonin | Anthocyanins | 9 | 9 | 9 | 2760000 | 2500000 | 3770000 | 1.363342 | 334444.4444 | 18.35141 |
| pmb0837 | Procyanidin A3 | Proanthocyanidins | 9 | 9 | 9 | 3160000 | 1610000 | 2300000 | 1.362756 | 261851.8519 | 17.99839 |
| pmb3074 | 3-O-p-Coumaroyl quinic acid | Quinate and its derivatives | 9 | 9 | 9 | 1450000 | 1170000 | 1030000 | 1.363438 | 135185.1852 | 17.04458 |
| pme3300 | Tricetin | Flavone | 9 | 9 | 9 | 2510000 | 311000 | 819000 | 1.355341 | 134814.8148 | 17.04062 |
| pmb0808 | 6,7-dihydroxycoumarin 6-O-quinic acid | Coumarins | 9 | 9 | 9 | 1450000 | 9 | 1460000 | 0.958286 | 107778.1111 | 16.7177 |
| pme1816 | Neochlorogenic acid (5-O-Caffeoylquinic acid) | Quinate and its derivatives | 9 | 9 | 9 | 773000 | 868000 | 1050000 | 1.363533 | 99666.66667 | 16.60482 |
| pmb0749 | O-Feruloyl quinic acid | Quinate and its derivatives | 9 | 9 | 9 | 173000 | 546000 | 1510000 | 1.355143 | 82555.55556 | 16.33308 |
| pmb3058 | Quinic acid O-glucuronic acid | Quinate and its derivatives | 33800000 | 30800000 | 29200000 | 9 | 9 | 9 | 1.363641 | 2.88E-07 | -21.7282 |
| pme0398 | Chlorogenic acid (3-O-Caffeoylquinic acid) | Quinate and its derivatives | 22800000 | 17200000 | 16900000 | 9 | 9 | 9 | 1.363516 | 4.75E-07 | -21.007 |
| pmb0639 | 8-C-hexosyl-apigenin O-hexosyl-O-hexoside | Flavone C-glycosides | 14500000 | 8060000 | 24600000 | 9 | 9 | 9 | 1.362386 | 5.73E-07 | -20.7362 |
| pma6647 | C-hexosyl-chrysoeriol O-hexoside | Flavone C-glycosides | 6340000 | 5840000 | 6310000 | 9 | 9 | 9 | 1.363675 | 1.46E-06 | -19.3854 |
| pmb0652 | C-hexosyl-apigenin O-pentoside | Flavone C-glycosides | 6300000 | 5470000 | 5240000 | 9 | 9 | 9 | 1.363606 | 1.59E-06 | -19.265 |
| pmb1108 | Luteolin 6-C-hexoside 8-C-hexosyl-O-hexoside | Flavone C-glycosides | 1530000 | 382000 | 11500000 | 9 | 9 | 9 | 1.346659 | 2.01E-06 | -18.9221 |
| pmb0623 | 6-C-hexosyl chrysoeriol O-hexoside | Flavone C-glycosides | 2360000 | 1200000 | 5440000 | 9 | 9 | 9 | 1.360537 | 0.000003 | -18.3466 |
| pma6516 | C-hexosyl-apigenin O-hexosyl-O-hexoside | Flavone C-glycosides | 2710000 | 1850000 | 2170000 | 9 | 9 | 9 | 1.363427 | 4.01E-06 | -17.9273 |
| pmb0629 | Chrysoeriol 6-C-hexoside | Flavone C-glycosides | 2590000 | 1940000 | 1210000 | 9 | 9 | 9 | 1.362641 | 4.70E-06 | -17.6977 |
| pmb4777 | 4-Hydroxy-7-methoxycoumarin-beta-rhamnoside | Coumarins | 584000 | 856000 | 851000 | 9 | 9 | 9 | 1.363425 | 1.18E-05 | -16.3727 |

100,540 unigenes for the samples, with an average transcript length of 1151 bp. The unigenes were mapped to the NR database and then applied to the Pfam database for annotation.

Using a false discovery rate (FDR) <0.01, log2FC>1 as threshold values, 4211 genes were found to be differentially expressed in PL vs. RG. Among them, 2035 genes were upregulated and 2176 genes were downregulated (Fig 4).

Differentially expressed genes potentially involved in anthocyanin biosynthesis were identified. These included *CHS, CHI, F3H, F3′H, F3′5′H, DFR, ANS* and *UFGT*.

The enriched GO (Gene Ontology) terms for DEGs were analysed (Fig 5). In biology process (BP) categories, the most significant terms were metabolic process, cellular process, and single-organism process. In molecular function (MF) and cellular component (CC) categories, binding and catalytic activity, and cell and cell part were the most enriched, respectively.

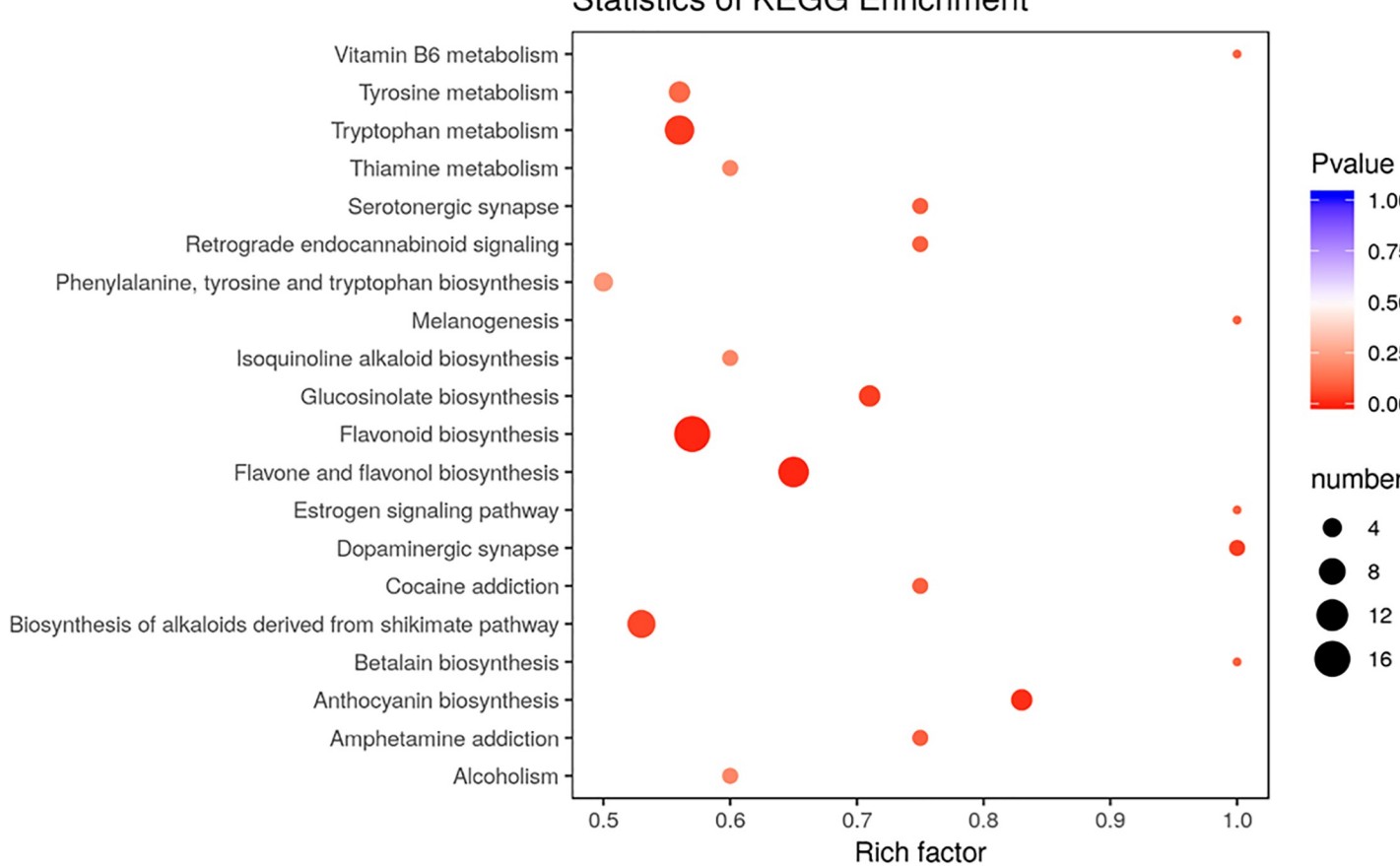

**Fig 2. KEGG pathway enrichment analysis based on the differentially accumulated metabolites in SYQ purple and green leaves.**

The identified genes were mapped to 105 KEGG reference pathways (Fig 6). These pathways included flavonoid biosynthesis (ko00941), purine metabolism (ko00230) and so on, and plant pathogen interaction (ko04626) and carbon metabolism (ko01200) had the highest enrichment factor values. In the flavonoid biosynthesis pathway, most genes were upregulated in purple leaves.

Transcription factors are essential regulators in anthocyanin biosynthesis. A total of 176 TFs were identified as either up- or downregulated between purples and green leaves. Of the most extensively studied TFs, one MYB and six bHLH were significantly different between purple and green leaves. MYB was highly expressed in purple leaves, four bHLH were upregulated, and two were downregulated.

## Network analysis of metabolites and transcripts in SYQ

The identified anthocyanins, their relevant compounds and the related genes were mapped onto their corresponding position in anthocyanin pathway. The results indicated that the compounds in this pathway were different in purple and green leaves. The abundances of the transcripts and composition of the compounds are shown via heatmap (Fig 7).

As shown in Fig 7, pelargonidin, dihydrokaempferol, and pelargonidin 3-O-glucoside were significantly accumulated in purple leaves, and dihydromyricetin, cyanidin, and cyanidin 3-glucoside had higher levels in green leaves. The majority of the transcripts were promoted in purple leaves, although the expression levels of *CHS* and *CHI* were inhibited. In our results,

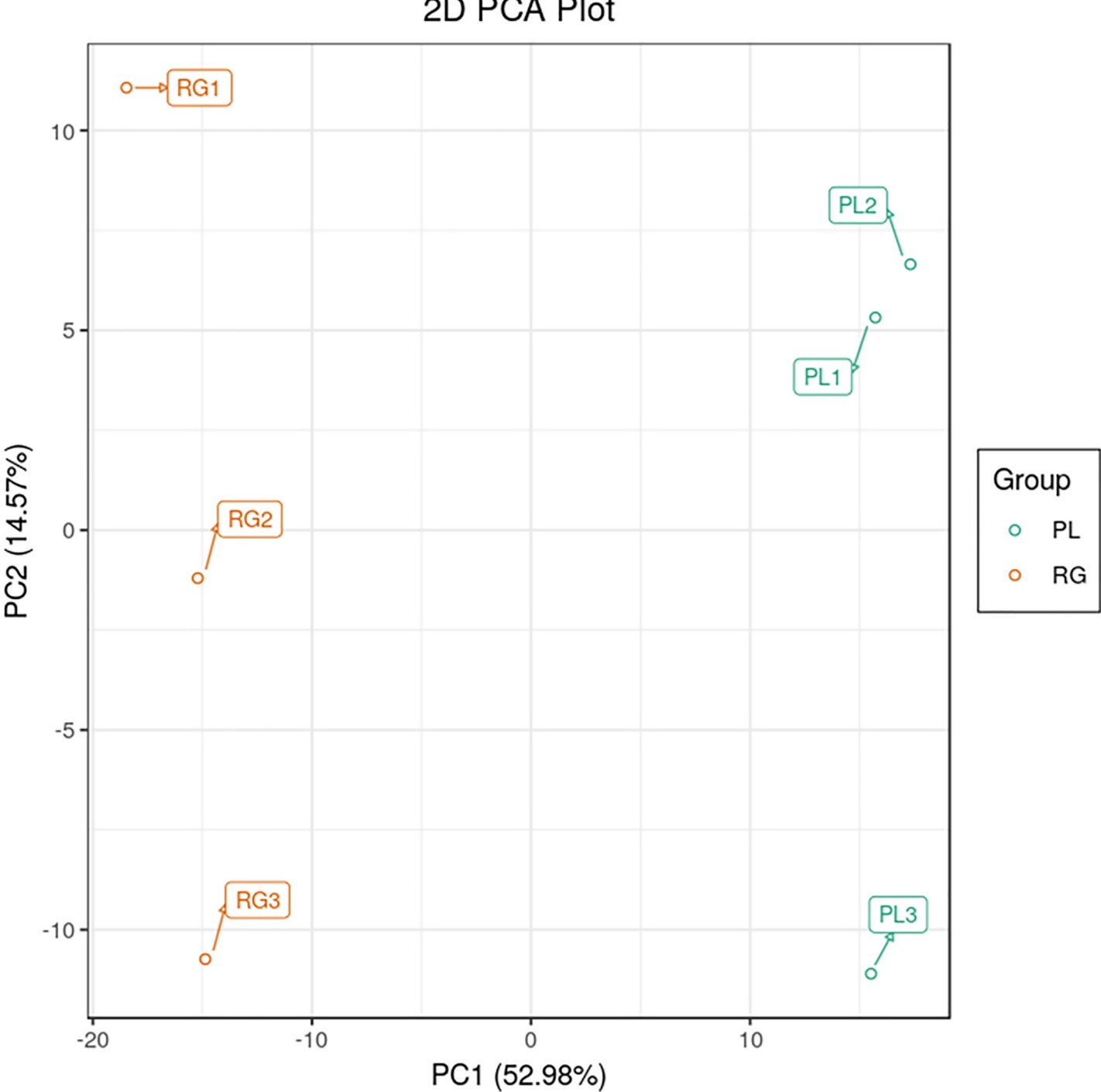

**Fig 3. PCA score plot of SYQ purple and green leaves.**

the expression levels of the transcripts and metabolites in the pelargonidin biosynthesis pathway were increased.

To gain a better understanding of the regulatory network of the leaf colour in the two accessions, the Pearson correlation test was performed for the metabolites and the transcripts. In total, 14 genes and 16 metabolites involved in flavonoid and anthocyanin pathways were

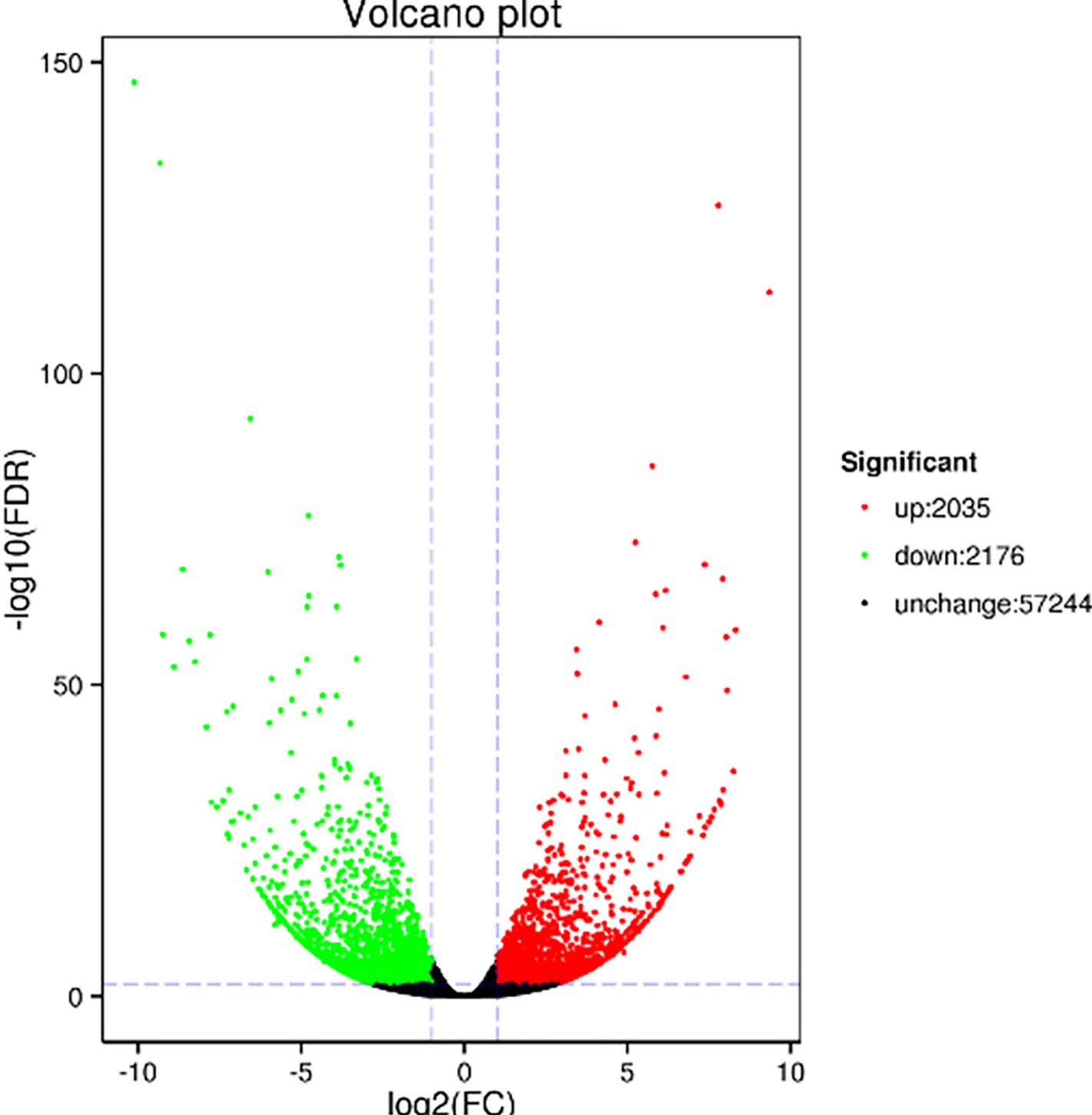

**Fig 4. Volcano plot of differentially regulated genes in PL vs RG.** The red dots are upregulated genes and green dots are downregulated genes.

subjected to Pearson correlation analysis (Fig 8). There were 141 significant correlation combinations between the genes and metabolites that had a Pearson correlation coefficient >0.8 and p value<0.05.

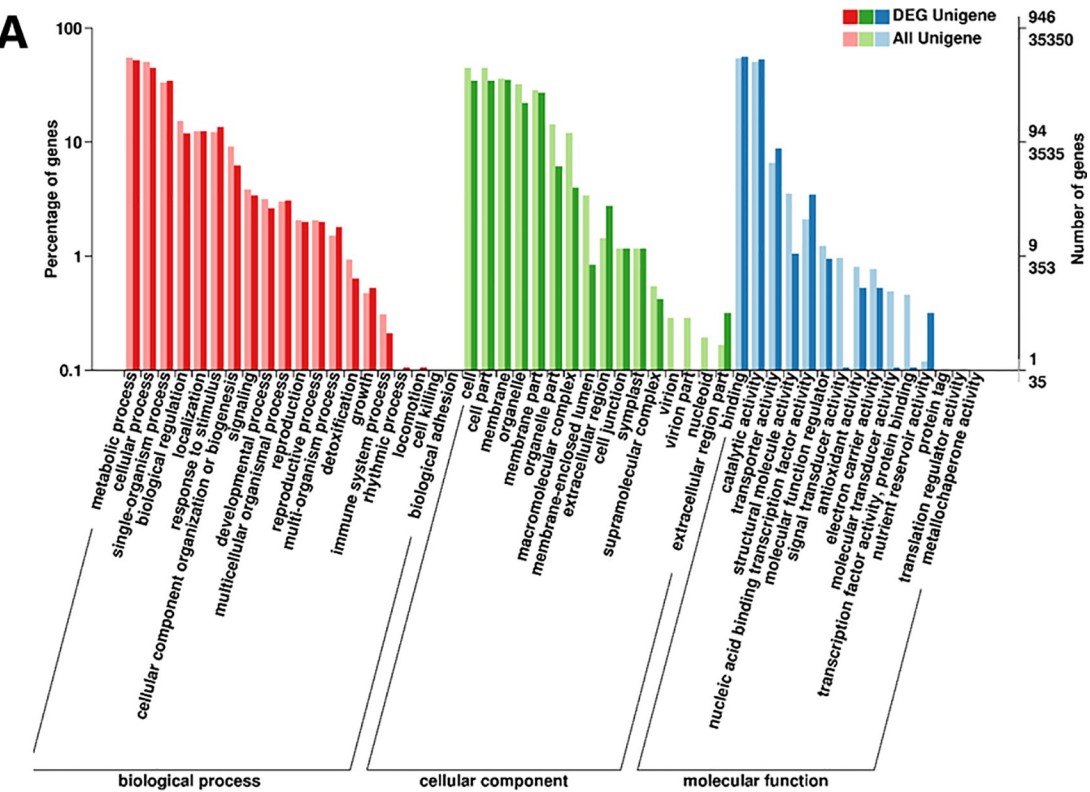

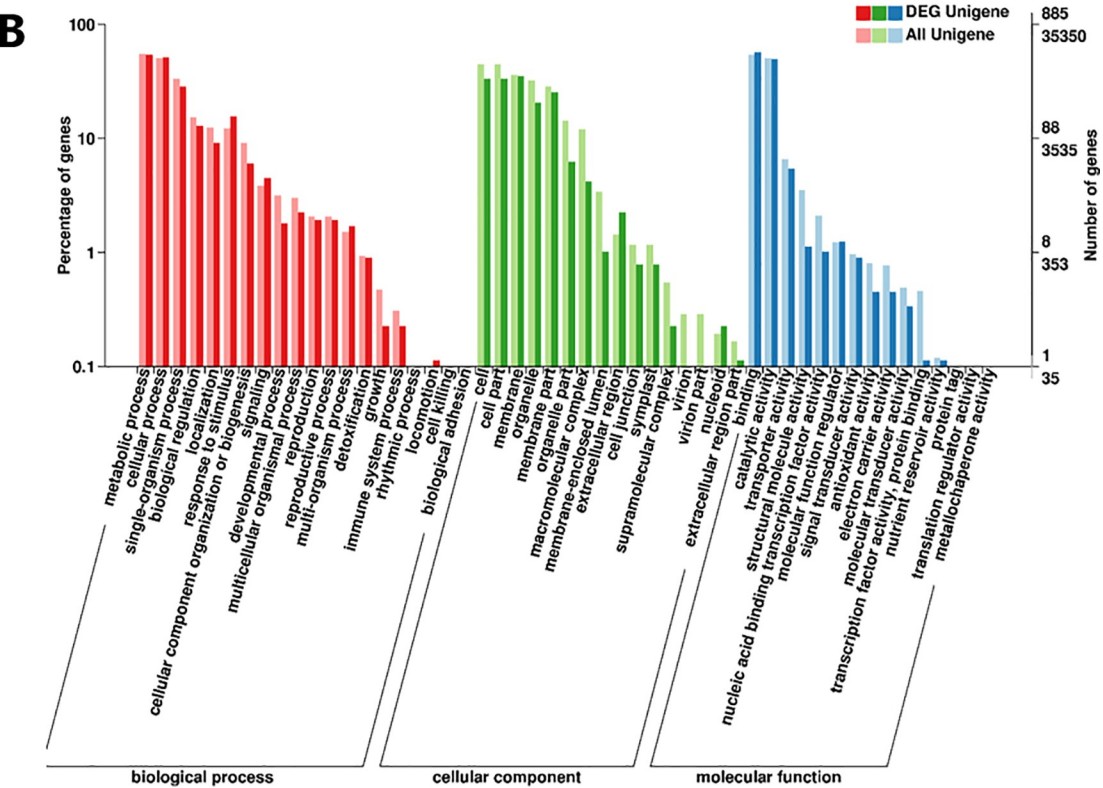

**Fig 5. GO classification of all DEGs in purple and green SYQ leaves.** Upregulated genes (A) and downregulated genes (B).

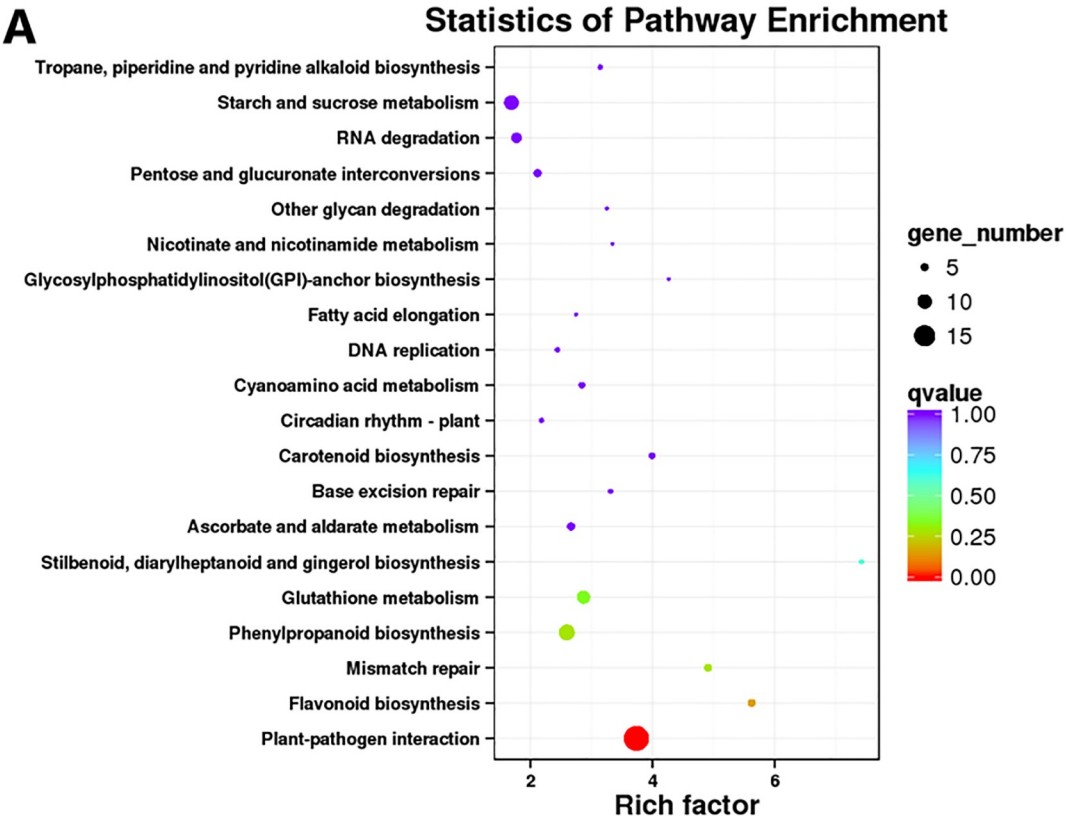

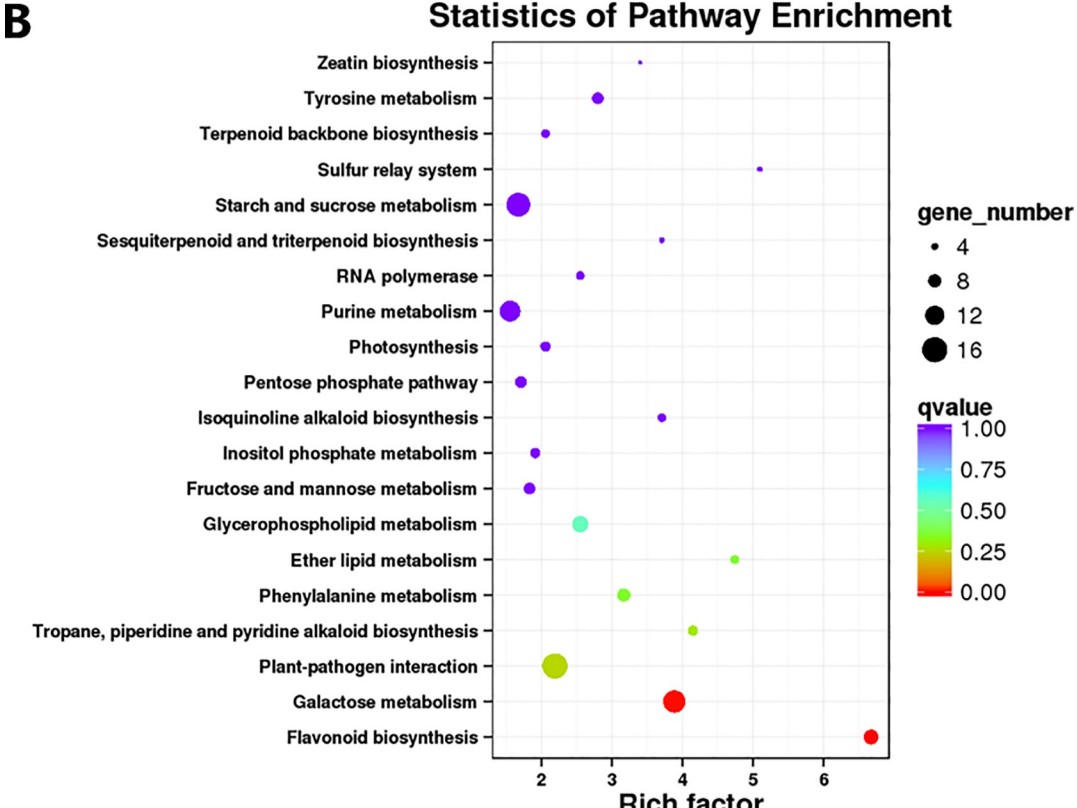

**Fig 6. KEGG pathway enrichment analysis based on all DEGs of SYQ purple and green leaves.** Upregulated genes (A) and downregulated genes (B).

The expression levels of the key genes in the anthocyanin biosynthesis pathway were analysed by real-time quantitative PCR (qPCR). As shown in Fig 9, *UGFT*, *ANS*, *DFR*, *F3'5'H*, and *F3H* accumulated at a much higher level in the purple leaves, whereas the expression of *CHI* was reduced in purple leaves. The expression pattern of the genes involved in the anthocyanin biosynthesis pathway revealed by qPCR was consistent with the RNA-seq results.

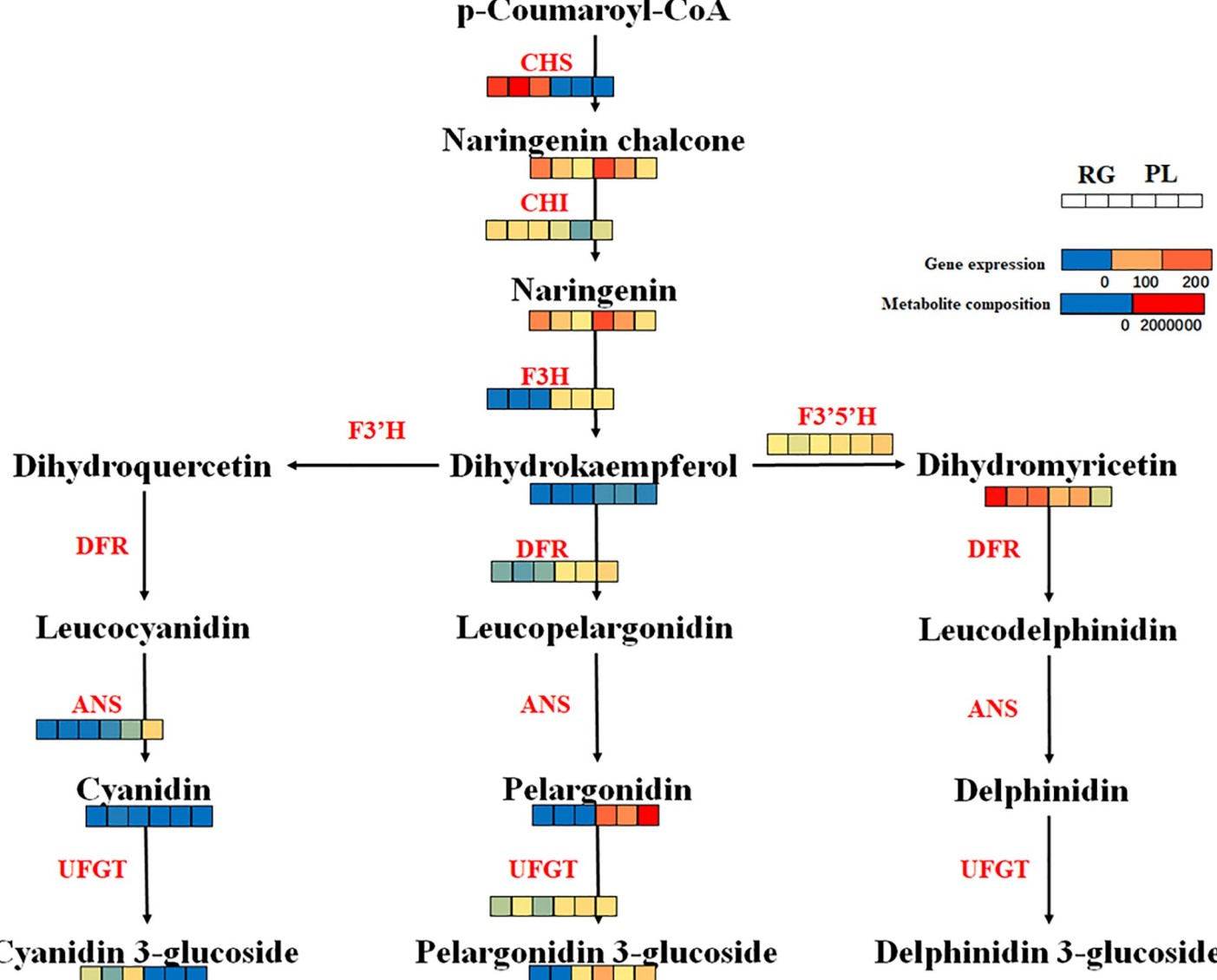

**Fig 7. Biosynthetic pathway of anthocyanins.** This pathway is constructed based on the KEGG pathway. Each coloured cell represents the normalized intensity of each compound ion according to the colour scale (three biological replicates×two cultivars, n = 6). The expression levels of chalcone synthase (CHS), chalcone isomerase (CHI), flavanone 3-hydroxylase (F3H), flavonoid 3'-hydroxylase (F3'H), flavonoid 3',5'-hydroxylase (F3'5'H), dihydroflavonol 4-reductase (DFR), anthocyanidin synthase (ANS) and flavonoid 3-O-glucosyltransferase (UFGT) are shown.

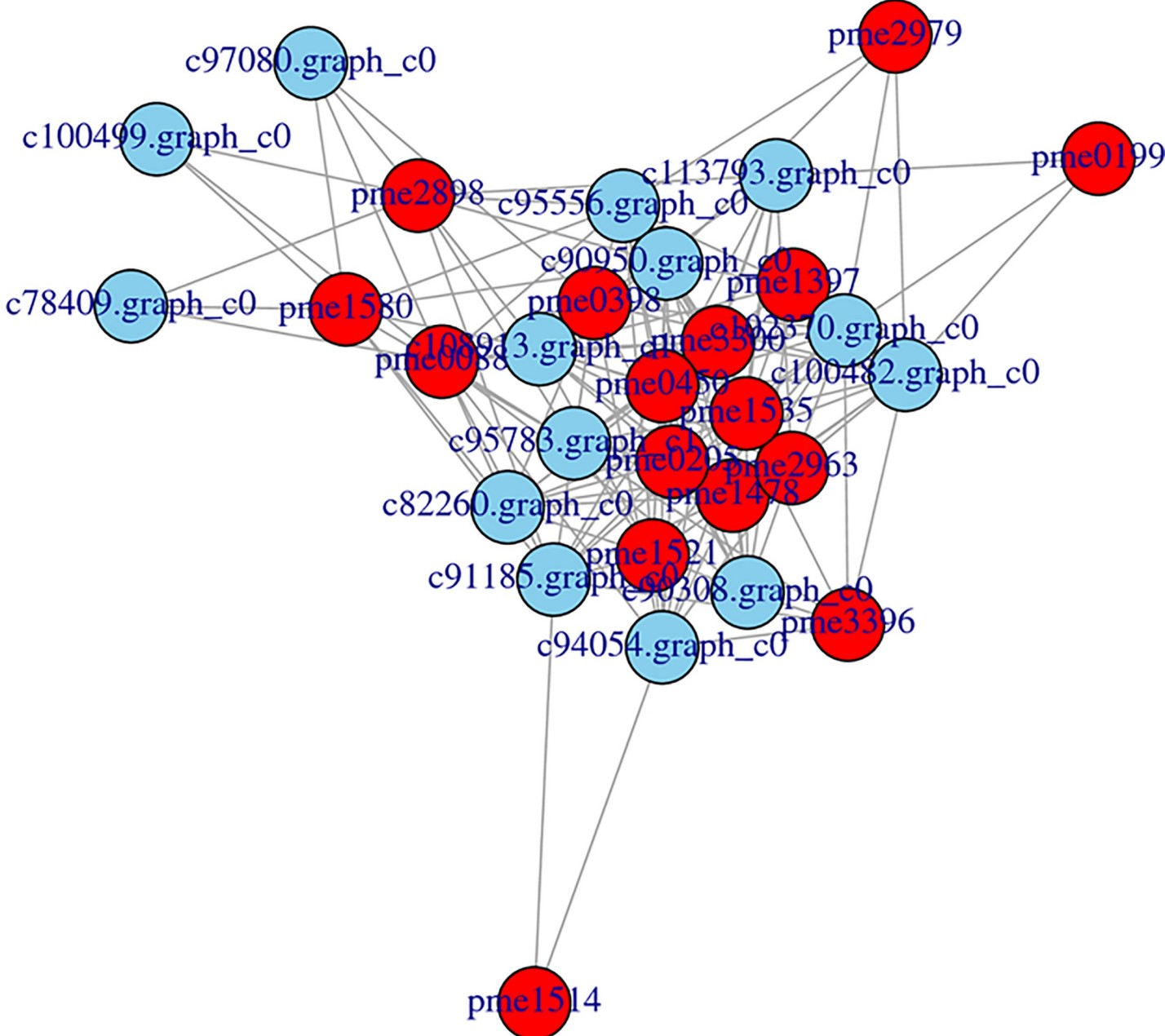

**Fig 8. Connection network between regulatory genes and anthocyanin-related metabolites.** The networks between metabolites and transcripts were visualized with Cytoscape software (version 2.8.2).

## Discussion

In this study, the differences in transcripts and metabolites in green and purple leaves were compared. Using ultra-performance liquid chromatography and tandem mass spectrometry, 597 metabolites were detected (Fig 1). As shown by hierarchical sample clustering, there was a clear separation between purple and green leaves (S2 Fig). In the differential accumulated metabolites, six metabolites annotated to anthocyanins were up-accumulated in purple leaves, while only two anthocyanin compounds were inhibited.

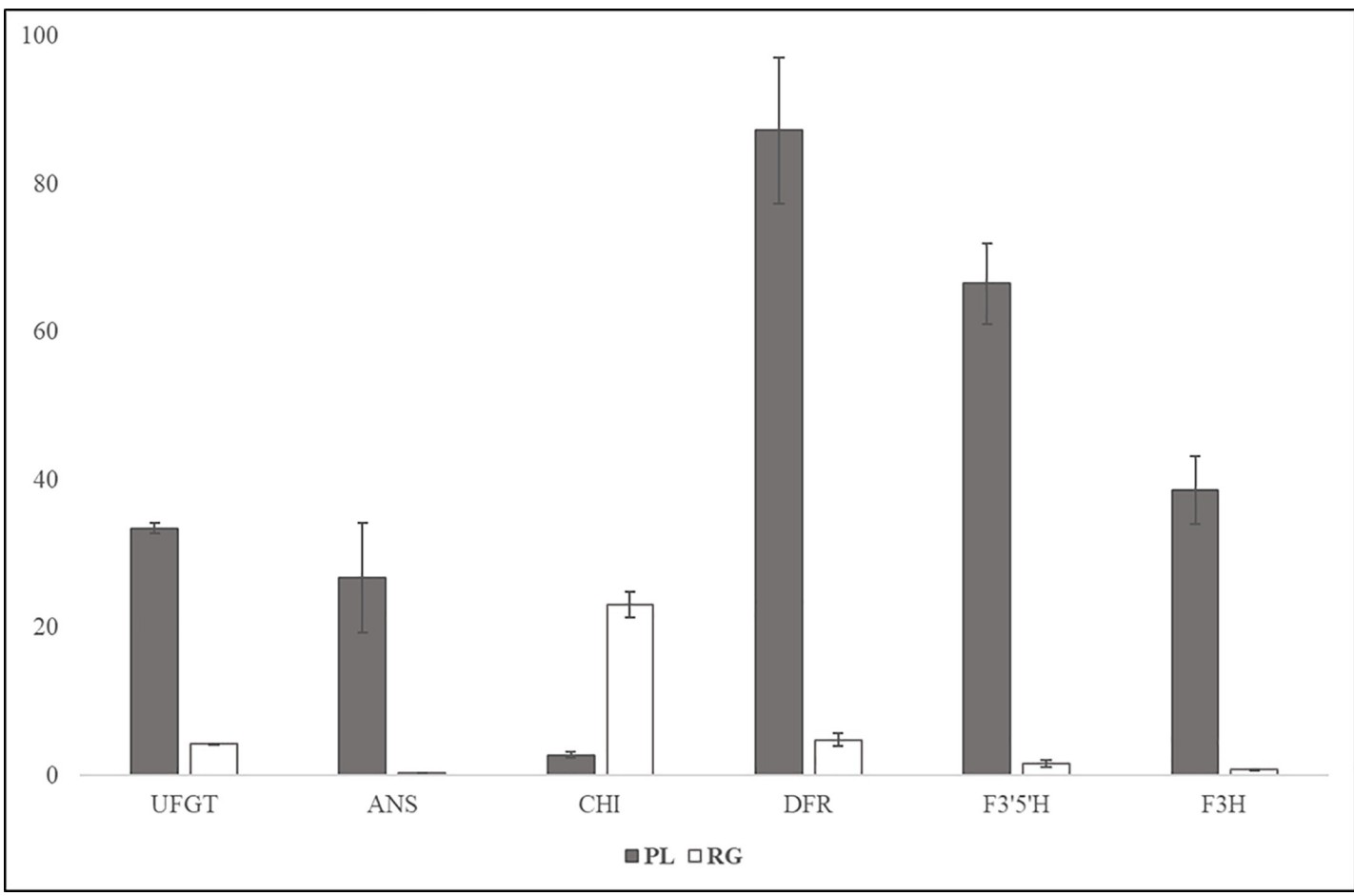

**Fig 9. qPCR analysis of anthocyanin biosynthesis-related genes in green- and purple-leaf SYQ cultivars.**

To investigate the genes regulating the differential pigmentation of SYQ leaves, transcriptome profiling was engaged. Anthocyanin accumulation corresponds to the expression of the genes involved in the biosynthesis pathway [5,18]. Our study showed that most of the key genes in the anthocyanin biosynthesis pathway were upregulated in purple leaves (Fig 7), indicating that the anthocyanin pathway was promoted in purple leaves. This finding was consistent with the gene expression results of the key genes in anthocyanin biosynthesis pathway (Fig 9). The largest log2FC (purple/green) value was found in *F3H*, which had a value of 5.869. F3H catalyses the conversion of flavanone into dihydroflavanol[19]. The upregulation of *F3H* facilitates the accumulation of anthocyanins.

There are many types of anthocyanins; cyanidin, delphinidin and pelargonidin have been recognized as the most common anthocyanins in plants. Different distribution of anthocyanins results in different leaf colours in fruits and leaves. For instance, the major type of anthocyanins in berries is cyanidin [20]; red pigment is generated from the accumulation of pelargonidin [21], and delphinidin makes plants look purple. In our results, the biosynthesis pathway of pelargonidin-3-glucoside was promoted at both the transcriptional and metabolic levels. The genes and metabolites work together to regulate the production of anthocyanins.

Transcription factors including MYB, bHLH and WD40 play vital roles in the regulation of flavonoid and anthocyanin pathway [22–24]. Only MYB had a higher expression level in purple leaves, and different bHLHs transcripts were differentially expressed in different leaves.

There are 8 WRKYs that showed higher expression levels in purple leaves. As suggested previously, WRKY may be involved in the regulation of anthocyanin biosynthesis. More anthocyanin was accumulated in the *WRKY* overexpressed lines compared to wild type in *Arabidopsis* [25].

Polysaccharides, phenolic acids and flavanols are the main compounds in SYQ with biological activities that can be used to cure some diseases. Although there are abundant bioactive compounds in SYQ root tubers [26], the shootsmay be more practical for use due to their easy access. In our results, the composition of the bioactive compounds in purple leaves was higher than in green leaves. Purple leaves may contain higher levels of antioxidant activity compared to green leaves. Therefore, purple leaves with higher anthocyanin contents are a superior resource for the improvement of SYQ quality.

Network analysis of the metabolites and transcripts was conducted. There were 141 significant correlation combinations between 14 genes and 16 metabolites. Transcript c90950. graph_c0 (CHS) was correlated with 14 metabolites, the most of those investigated (Fig 8). CHS is the initial enzyme of flavonoid biosynthesis; it catalyses the synthesis of naringenin chalcone from 4-coumaroyl-CoA and malonyl-CoA. In different plant species, the expression pattern of CHS of red tissues and green tissues are different [27]. The expression level of *CHS* was found to be significantly upregulated in red tomato fruits, while *CHS* expression was similar between pigmented and non-pigmented tomato mutants [28]. Data obtained indicated that the metabolites and genes work co-ordinately to regulate anthocyanin biosynthesis. Regulatory genes that were highly correlated with the accumulation of metabolites were identified; these could provide new insights into the regulatory mechanism of anthocyanin biosynthesis.

## Conclusions

We analysed the regulatory network of anthocyanin biosynthesis integrating the metabolome and transcriptome in the purple and green leaves of SYQ. Correlation analysis of the metabolites and transcripts involved in anthocyanin biosynthesis pathway was conducted. A regulatory network of metabolites and transcripts related to leaf colour could be a resource for the exploitation of the mechanism of anthocyanin regulation in SYQ.

## Supporting information

**S1 Fig. Phenotypes of purple-leaf SYQ (A-PL) and green-leaf SYQ (B-RG).** Red and yellow arrows indicate the leaves shown in S1 Fig. Yellow arrows indicate the position where the leaves were collected for further experiments.
(TIFF)

**S2 Fig. Hierarchical clustering of differentially accumulated metabolites in PL and RG sample.** Normalized contents of different compounds are represented as colours ranging from green (-2) to red (2).
(TIFF)

**S1 Table. Sequence information for primers.**
(XLSX)

**S2 Table. Gene accession numbers and sequences of DEGs.**
(XLSX)

## Author Contributions

**Data curation:** Jianli Yan, Lihua Qian.

**Formal analysis:** Jieren Qiu.

**Funding acquisition:** Jianli Yan, Songlin Ruan.

**Investigation:** Jianli Yan, Weidong Zhu, Qiujun Lu, Xianbo Wang, Qifeng Wu, Yuqing Huang.

**Methodology:** Jianli Yan, Lihua Qian, Weidong Zhu, Jieren Qiu, Qiujun Lu, Xianbo Wang, Qifeng Wu, Yuqing Huang.

**Resources:** Lihua Qian, Weidong Zhu, Qifeng Wu.

**Supervision:** Qiujun Lu.

**Validation:** Jieren Qiu.

**Visualization:** Xianbo Wang.

**Writing – original draft:** Yuqing Huang.

**Writing – review & editing:** Songlin Ruan.

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
