## [Decision Letter · Decision Letter 0]

2 Jan 2020

PONE-D-19-31044

Integrated analysis of transcriptome and metabolome reveals alterations in expression of genes involved in anthocyanin biosynthesis in purple and green leaves of Tetrastigma hemsleyanum

PLOS ONE

Dear Mr. Ruan,

Thank you for submitting your manuscript to PLOS ONE. After careful consideration, we feel that it has merit but does not fully meet PLOS ONE’s publication criteria as it currently stands. Therefore, we invite you to submit a revised version of the manuscript that addresses the points raised during the review process.

We would appreciate receiving your revised manuscript by Feb 16 2020 11:59PM. To enhance the reproducibility of your results, we recommend that if applicable you deposit your laboratory protocols in protocols.io, where a protocol can be assigned its own identifier (DOI) such that it can be cited independently in the future. For instructions see: http://journals.plos.org/plosone/s/submission-guidelines#loc-laboratory-protocols

We look forward to receiving your revised manuscript.

Kind regards,

Zhong-Hua Chen, Ph.D.

Academic Editor

PLOS ONE

Journal Requirements:

3. In the Introduction to your manuscript, please remove any claims regarding the medicinal benefits of Tetrastigma hemsleyanum extracts which are not substantiated by robust scientific and clinical evidence.

Reviewers' comments:

Reviewer's Responses to Questions

**Comments to the Author**

1. Is the manuscript technically sound, and do the data support the conclusions?

Reviewer #1: Yes

Reviewer #2: Yes

2. Has the statistical analysis been performed appropriately and rigorously? 

Reviewer #1: Yes

Reviewer #2: Yes

3. Have the authors made all data underlying the findings in their manuscript fully available?

Reviewer #1: Yes

Reviewer #2: Yes

4. Is the manuscript presented in an intelligible fashion and written in standard English?

Reviewer #1: Yes

Reviewer #2: Yes

5. Review Comments to the Author

Reviewer #1: Please modify the manuscript according to the revised MS.

The study described in their manuscript, authors explored the regulatory networks of anthocyanin biosynthesis from the metabolome and transcriptome, the plant materials were very interesting with the green and purple leaves. Transcript and metabolite profiles were integrated to find the differential metabolites and genes, and their correlation for anthocyanin biosysthesis.

The author should discribe the background of the two accessions of SYQ, and their ralation.

I comment on related issues and hope the authors find these helpful to polish their work before final publication.

Reviewer #2: This manuscript applied metabolome and transcriptome to reveal the alteration of regulatory network in anthocyanin biosynthesis in SYQ. It is meaningful to discover some novel genes and metabolites for the improvement of anthocyanin content and superior cultivars. However, there are some problems on writing of this manuscript.

6. PLOS authors have the option to publish the peer review history of their article (what does this mean?). If published, this will include your full peer review and any attached files.

Reviewer #1: No

Reviewer #2: No

---

## [Author Response · Author response to Decision Letter 0]

22 Jan 2020

Responses to reviewers’ comments and suggestion

First of all, we are deeply indebted to your careful examination and valuable comments for further improvement of this manuscript. Accordingly, we revised the MS according to your comments and suggestion. The detail revisions are as follows.

From Vicky Stabler:

1.Please ensure that your manuscript meets PLOS ONE's style requirements, including those for file naming.

Au:Thank you for your comment.We have re-checked the style andfile naming of our manuscript.

2.We suggest you thoroughly copyedit your manuscript for language usage, spelling, and grammar. If you do not know anyone who can help you do this, you may wish to consider employing a professional scientific editing service. 

Au: We employed American Journal Experts (AJE) to polish our manuscript. The new version was submitted.

3.In the Introduction to your manuscript, please remove any claims regarding the medicinal benefits of Tetrastigmahemsleyanum extracts which are not substantiated by robust scientific and clinical evidence.

Au: Relevant information was deleted in the revised version.

4.In your Data Availability statement, you have not specified where the minimal data set underlying the results described in your manuscript can be found.

Au: A new supplementary table (Table S2) of the sequences of all DEGs was added.

5.PLOS requires an ORCID iD for the corresponding author in Editorial Manager on papers submitted after December 6th, 2016. 

Au: The ORCID iD was validated in the submission system.

Reviewer #1: Please modify the manuscript according to the revised MS.

1.The author should describe the background of the two accessions of SYQ, and their relation.

Au: Relevant information was added in the revised version.

2. I comment on related issues and hope the authors find these helpful to polish their work before final publication.

Au: Thank you for your careful examination and valuable comments. In the revised version, the spelling and grammatical errors were corrected.

Reviewer #2: This manuscript applied metabolome and transcriptome to reveal the alteration of regulatory network in anthocyanin biosynthesis in SYQ. It is meaningful to discover some novel genes and metabolites for the improvement of anthocyanin content and superior cultivars. However, there are some problems on writing of this manuscript.

-Minor issues

1. Line 35-36, “Previous findings have demonstrated that the principal and functional components of SYQ, such as flavonoids, polysaccharide and polyphenol, have various effect on anti-cancer, liver protection, and anti-inflammatory”, I think polyphenol include flavonoids.

Au:Thank you for your comment. It was corrected in revised version.

Line 40, “Anthocyanins also involved in biotic and abiotic stress defending responses”, rephrase this sentence.

Au: We rewrote it in the revised version.

Line 43-44, “The color of the plants leaves is mainly determined by the composition and amount of the anthocyanins, the anthocyanins are the products of the branched flavonoid biosynthesis pathway”, “amount” is not proper in this sentence.

Au:Sorry we made an error. “amount” was changed to “concentration” in the revised version.

Line 50, “The transcripts in low abundance or some unrevealed ever can be identified”, rephrase this sentence.

Au: This sentence was rewritten in revised version.

Line 51-52, “Advanced in metabolomics with the help of liquid chromatography / mass spectrometry (LC/MS) has been a powerful tool to discover a large number of compounds compared with traditional chemical analysis”, “Advanced in metabolomics”? Rephrase this sentence. 

Au:Thank you for your comment. “Advanced” was changed to “Advances” in the revised version.

Line 203, check and rephrase this sentence.

Au: Relevant correction was made in revised version.

Line 220-221, “Although the bioactive compounds in root tubers of SYQ is abundant, the aerial parts of SYQ may be more practical due to its easily access”, what are the aerial parts? And why are these parts easily access? 

Au: “aerial parts” was changed to “shoots” in the revised version.

Line 223-224, check and rephrase this sentence. 

Au: The correction was made in the revised version.

---

## [Decision Letter · Decision Letter 1]

24 Feb 2020

Integrated analysis of the transcriptome and metabolome of purple and green leaves of Tetrastigma hemsleyanum reveals gene expression patterns involved in anthocyanin biosynthesis

PONE-D-19-31044R1

Dear Dr. Ruan,

We are pleased to inform you that your manuscript has been judged scientifically suitable for publication and will be formally accepted for publication once it complies with all outstanding technical requirements.

With kind regards,

Zhong-Hua Chen, Ph.D.

Academic Editor

PLOS ONE

Additional Editor Comments (optional):

Reviewers' comments:

Reviewer's Responses to Questions

**Comments to the Author**

1. If the authors have adequately addressed your comments raised in a previous round of review and you feel that this manuscript is now acceptable for publication, you may indicate that here to bypass the “Comments to the Author” section, enter your conflict of interest statement in the “Confidential to Editor” section, and submit your "Accept" recommendation.

Reviewer #1: All comments have been addressed

Reviewer #2: All comments have been addressed

2. Is the manuscript technically sound, and do the data support the conclusions?

Reviewer #1: Yes

Reviewer #2: Yes

3. Has the statistical analysis been performed appropriately and rigorously? 

Reviewer #1: Yes

Reviewer #2: Yes

4. Have the authors made all data underlying the findings in their manuscript fully available?

Reviewer #1: Yes

Reviewer #2: Yes

5. Is the manuscript presented in an intelligible fashion and written in standard English?

Reviewer #1: Yes

Reviewer #2: Yes

6. Review Comments to the Author

Reviewer #1: (No Response)

Reviewer #2: (No Response)

7. PLOS authors have the option to publish the peer review history of their article (what does this mean?). If published, this will include your full peer review and any attached files.

Reviewer #1: No

Reviewer #2: No

---

## [Editor Report · Acceptance letter]

26 Feb 2020

PONE-D-19-31044R1 

Integrated analysis of the transcriptome and metabolome of purple and green leaves of *Tetrastigma hemsleyanum* reveals gene expression patterns involved in anthocyanin biosynthesis 

Dear Dr. Ruan:

I am pleased to inform you that your manuscript has been deemed suitable for publication in PLOS ONE. Congratulations! Your manuscript is now with our production department. 

With kind regards,

on behalf of

Dr. Zhong-Hua Chen 

Academic Editor

PLOS ONE